# A Novel Multi-Criteria Sorting Model Based on AHP-Entropy Grey Clustering for Dealing with Uncertain Incoming Core Quality in Remanufacturing Systems

**Mohamad Imron Mustajib [1,2,*]** , **Udisubakti Ciptomulyono [1,*]** and **Nani Kurniati [1]**

[1]  Department of Industrial and System Engineering, Institut Teknologi Sepuluh Nopember, Surabaya 60111, Indonesia; nanikur@ie.its.ac.id
[2]  Department of Industrial and Mechanical Engineering, Universitas Trunojoyo Madura, Bangkalan 69162, Indonesia
[*]  Correspondence: imronmustajib@gmail.com or imronmustajib@trunojoyo.ac.id (M.I.M.); udisubakti@ie.its.ac.id (U.C.)

**Abstract:** Remanufacturing is a key pillar of a circular economy and helps in recovering used products by extending their life cycle via remanufacturing them into new products. A vital aspect in a remanufacturing system is the quality assessment of incoming worn-out products (cores) prior to remanufacturing to ensure that non-conforming cores are discarded at an early stage in order to avoid unnecessary processing. Therefore, quality sorting plays an important role in core acquisition for remanufacturing systems when attempting to mitigate uncertain incoming core quality as an immediate solution. The main problem is that it is difficult to acquire the important information required to decide on the sorting of incoming cores, such as the core quality. The data are also commonly limited, not always available, or inaccurate. Grey systems are powerful methods in decision making when handling uncertainty with small data. In this paper, we consider the usefulness of grey systems for handling uncertain quality information for sorting incoming cores in a remanufacturing system. For this reason, we propose a multi-criteria quality sorting model based on an analytical hierarchy process (AHP)-entropy model that is coupled with grey clustering using possibility functions. The quality criteria for sorting the incoming cores are considered according to the technological, physical, and usage conditions. To demonstrate the practical contribution of this research, a case study of the quality sorting problem with a heavy-duty equipment remanufacturer is presented. The proposed model consistently classifies the quality of used hydraulic cylinders into two grey classes.

**Keywords:** core acquisition; quality grading; grey decision making; analytical hierarchy process

## 1. Introduction

In the new global economy, the concept of a circular economy has become a central issue of international concern. Many studies by researchers and policymakers in recent years have focused on circular economies as possible solutions to pursue the global issue of sustainable development goals. There are multiple definitions of a circular economy. In general terms, a circular economy can be viewed as a closed-loop industrial system that has activities for the reduction, reuse, and recycling of resources for the purpose of developing sustainability. The feedback loops indicate that recycling and recovery are the main concerns in a circular economy [1]. The strategy of a circular economy is to retain services in use for as long as possible in order to gain maximum value from them during use, and then to recover and restore them at the end of any lifetime of service [2].

Remanufacturing is the backbone of the circular economy, which helps in salvaging used products via recovery strategies to extend product life cycles. Such processes start with the acquisition of cores to be remanufactured into new products. The remanufactured products should fulfill a performance that is at least equal to the original products as a

representation of customer requirements. The three main activities in any remanufacturing system include core acquisition, remanufacturing operations, and re-marketing. Core acquisition is challenging for remanufacturers as remanufacturing in closed-loop supply chains is characterized by uncertainty for the quality, time, and volume of returns. The term uncertainty is generally understood to mean lack or incompleteness of information, i.e., known only as incompletely or imprecisely. Incompleteness is characterized as a measure of an inexistence of basic source data that should be provided to complete a given business process of remanufacturing. As a result, uncertainty leads to economic (cost and feasibility) and technical (remanufacturability, scheduling, and process planning) risks for remanufacturing companies. Therefore, these uncertainties need to be mitigated to achieve the desired performance.

The quality of a product is concerned with the degree to which the requirements are fulfilled by a set of key characteristics. Quality variation of a new material is one of the most significant factors causing non-conformance to the requirements for final product quality in manufacturing systems. The important factor of the incoming core quality in remanufacturing system is comparable to the new material quality in manufacturing system. Quality is an essential criterion for the performance evaluation of used products and is a key driver in remanufacturing decisions. The quality of incoming cores has a significant effect on the remanufacturing cost and cost of quality. Whenever the sorting of incoming cores is possible, it may lead to easy remanufacturing planning for minimizing these costs. If the quality level of incoming cores is better, remanufacturing costs will be low, and any used products can be saved at a value that increases with their level of quality. In order to return the used products into a good functional state, the highest quality cores may need a limited reconditioning process (thus entailing lower process costs), whereas the worst quality cores will need comprehensive processes or part replacements [3]. An example of this can be seen from study carried out by Behdad and Thurston [4], in which they formulated a model to deal with the issue of upgrading a used part and bringing it to a desired quality level at the end of its life by considering the current quality levels of the parts. The evidence of the important role of quality inspection and sorting also can be seen in the case of the decision-making process for upgrade planning, in which the incoming products are classified into several different quality classes upon arrival, and shortly after they are transferred to the remanufacturing unit [5]. A quantitative study by Ferguson et al. [6] described the finding that over a wide range of parameter values typically seen in the remanufacturing industry, a grading system increases profit and consequently profit increases as the amount of returns increases. Therefore, sorting and quality grading play major roles in core acquisition for remanufacturing systems when considering the management of uncertain incoming core quality. Sorting operations in core acquisition are vital for two important reasons. First, to identify the physical, usage, and technological conditions of incoming cores, where they are sorted by their quality level before any remanufacturing processes. Second, sorting operations represent an immediate solution to mitigating quality uncertainty in the core acquisition [7–9].

Sorting problems can be broadly defined as a case where a sets of alternatives are grouped in an ordinal manner according to the absolute evaluation, beginning with those that include the most preferred alternatives to those that include the least preferred alternatives [10]. Quality sorting in a remanufacturing system plays an important role for grading cores according to their different conditions in order to plan the remanufacturing process and the reduce the cost of remanufacturing. Cores with similar quality grades can be remanufactured with dedicated processes such that the time and cost can be handled efficiently. Quality classification and sorting policies are urgent and direct solutions that are used in remanufacturing systems to handle the source variability in incoming products [8]. Unfortunately, a lack of information from end users can lead to inaccuracy in the quality sorting process. Complete information is needed in order to determine the quality levels of incoming cores. Moreover, when cores are sorted into different quality grades, limited

quality data makes it difficult to estimate the associated quality level and the process planning stage thus becomes difficult.

To manage quality uncertainty issues, the quality control of incoming cores has become an important strategy for remanufacturing companies. The term quality control is used here to refer to the part of core acquisition management focused on fulfilling quality requirements for incoming cores. Recent evidence suggests that fast sorting in long-term quality control can be accomplished by installing information and communication technology, e.g., automatic sorting systems using radio frequency identification (RFID) or semi-automatic, sensors, bar codes, and other technologies to automate product monitoring and testing in order to acquire valuable usage data to assess remanufacturing feasibility [5,10,11]. For instance, in order to record operating hours and speed, Bosch integrated chips into electric motor power tools [12]. Afterward, they evaluated tool quality and sorted into two classes, either remanufacturable or non-remanufacturable. Unfortunately, these methods do not always guarantee usefulness for many remanufacturers. These technologies are only useful for original equipment remanufacturers (OER) who have control over the product design and wish to use such practice to invest in long-term payback if it is economically feasible. These methods are an impractical option for independent or contract remanufacturers.

Alternative approaches are necessary to solve the problem of uncertain quality for remanufacturing with short-term quality control. Quality sorting is used to directly control and categorize incoming cores into several quality classes, from the most preferred class to the less preferred class to justify the economic and technical viability for remanufacturing. One can expect that cores with a similar quality class will need similar re-processing operations. In this context, quality uncertainty refers to the required information to be used to assess which conditions of the incoming core are not precisely known. Several methods currently exist to deal with uncertainty. A well-known example is the probability for randomness behavior which is based on a density function, and fuzzy sets for fuzzy problems according to a membership function for ambiguity. The key differences between probabilistic and fuzzy sorting are that precise classifications for the number of alternatives each class has is given by fuzzy sorting and the percentage chances that an alternative is held by each class is given by probability sorting [13]; however, the probability statistic approach requires large sample data quantities to determine the probability density function. By way of illustration, Gavidel and Rickli [14] showed how large data quantities were required for triage as an agile sorting strategy in extreme core arrival scenarios. Obtaining large data quantities can be time-consuming and is often technically difficult to perform. On the other hand, the fuzzy mathematical method depends on experience and cognitive aspects to develop a fuzzy membership function.

The quality sorting problem for incoming cores can be subjectively or objectively uncertain depending on the facts. For example, the classification of physical condition based on damage level as determined by visual inspection is subjective, as it can change from one inspector to another. In contrast, the classification of cores according to their usage conditions according to the frequency of use is objective. As a result, there remains a need for an efficient method that can handle quality uncertainty with small data quantities in the sorting problem of incoming cores for remanufacturing. Quality sorting problems with incoming cores may have multiple and conflicting criteria as per the technological, physical, and usage conditions. To overcome the conflicting criteria, a multi-criteria decision-making methodology was constructed using an analytical hierarchy process (AHP). The primary benefit of the use of an AHP is that it can be used for criteria which are quantitative and qualitative, and additionally the capacity to consider the subjective views of decision-makers.

In 1982, grey system theory (GST) was first proposed by Professor Julong Deng from Huazhong University of Science and Technology as a quantitative model for limited and incomplete data [15]. Incompleteness and inadequacy of data is a basic feature of uncertain systems [16]. GST has been widely used in various fields for decision-making problems with uncertainty. This paper sets out to investigate the usefulness of grey systems for

handling uncertain quality information for sorting incoming cores in a remanufacturing system. The concern with grey system theory is that uncertainty issues arising from limited or missing data are difficult to address with probability theory.

In grey systems, grey sets employ the basic concept of grey numbers and deal with the characteristic function values of a set as grey numbers. A grey number is a number that has clear upper and lower limits, but which has an unknown location within the limits. Moreover, grey clustering is a method that may be defined as the branch of grey system theory, which is concerned with the classification of observation indices or observation objects into definable groups using grey incidence matrices or grey possibility functions [17]. Grey clustering evaluation models using possibility functions have been extensively studied for uncertain systems analysis. If a grey number's value data are known to some degree, we may use the possibility function to explain the possibility of the potential values that should be taken up by the grey number. Many recent studies [18–20] have shown that a method based on grey clustering was helpful for classification problems in remanufacturing operations with uncertain conditions because grey classification has moderate computation complexity. Difficulties arise when an attempt is made to consider multi-level criteria when making decisions regarding grey sorting problems. Moreover, the current major challenges in quality sorting are addressed in the research questions listed in Table 1. The table also provides relevant research objectives.

**Table 1.** Relevant research questions for research objectives.

| | Research Questions | | Research Objectives |
|---|---|---|---|
| a. | How does one overcome the sorting problem when the incoming cores feature uncertain quality? | a. | To develop a multi-criteria sorting model with uncertain quality. |
| b. | What criteria should be considered in the quality evaluation of incoming cores? | b. | To identify and propose the criteria for the quality evaluation of incoming cores. |
| c. | How does one evaluate the relative importance among the quality criteria with incomplete data? | c. | To determine the weights of the considered criteria. |
| d. | How does one assign the set of incoming cores into pre-determined quality classes? | d. | To establish the effective method for cores assignment associated with the sorting methods |

This paper proposes an effective sorting model suitable for core quality classification. In the present study, we propose the use of an AHP for structuring the problem in addition to Shannon entropy and grey clustering for handling quality uncertainty. The clustering weights could be found through different methods with subjective and objective weighting within multiple quality criteria. Subsequently, the development of the model will be accomplished by assigning the incoming cores into different quality classes for remanufacturing. The rest of this paper is organized into the following sections. Following the above introduction, Section 2 presents a literature review. Subsequently, Section 3 describes the conceptualization and construction of the sorting model by combining grey clustering and the AHP approach. A case study is presented in Section 4 to demonstrate the application of the proposed model for the problem of the quality sorting of incoming cores in heavy-duty equipment parts. Multi-criteria sorting for incoming cores into predefined quality classes based on their dominant attributes is proposed in this section. The results, findings, and managerial insights are also included in this section. Finally, conclusions and further areas for study are presented in Section 5.

## 2. Literature Review

Researchers and practitioners have recognized uncertainty as one of the most challenging elements in making decisions regarding remanufacturing [21]. The term "uncertain" has been used to refer to situations in which something is not known or something is not

complete or not certain. Such situations may include physical measurements or unknown variables [22]. Table 2 lists several techniques that have been developed to deal with distinct kinds of uncertainties. The use of probability statistics is one of the most common methods for investigating uncertainty conditions.

**Table 2.** Comparative overview of the approaches for modeling uncertainty [17].

| Uncertainty Approaches | Research Objects | Primary Set | Description Method | Procedure | Data Prerequisite | Objective | Data Availability |
|---|---|---|---|---|---|---|---|
| Probability statistics | Stochastics | Cantor set | Density function | Frequency | Known distribution | Historical law | Large |
| Fuzzy set system | Cognitive | Fuzzy set | Membership function | Cut set | Known membership | Cognitive expression | Based on experience |
| Rough set | Boundary | Approximate set | Upper and lower | Dividing | Equivalent relationship | Approximation | Information form |
| Grey system theory | Poor information | Grey number set | Possibility function | Sequence operator | Possibility function | Law of reality | Small |

Unfortunately, such methods require large sample data quantities based on historical law for determining probability density function. Fuzzy systems are the most popular methods for investigating subjective uncertainty in terms of vagueness or ambiguity in linguistic statements. Fuzzy systems concern the study of cognitive uncertainty issues, where the research objects have the characteristics of clear and unclear information. Different from fuzzy sets, a rough set approximates a set of numbers using two definable upper and lower sets. In terms of data granularity (referring to the quality of information), a rough set is described and a finer granularity provides a more accurate approximation [15]. Meanwhile, the grey system approach has many attractive features, notably, to support for any data distribution requirement with a small data size. Further, compared to other methods applied to handle uncertainty, grey systems do not require extensive data and are more pragmatic and efficient in complex systems and require fewer mathematical calculations [23].

The following presents a brief description of the literature analysis regarding uncertainty in remanufacturing systems. It can be seen in Figure 1 that the front end in remanufacturing systems is the acquisition of worn-out products (usually referred to as cores). Product acquisition acts as a gatekeeper in remanufacturing for obtaining used products or parts from end users/customers for further reprocessing. This task affects all remanufacturing operations but is complicated by the high uncertainty about the quality, timing, and quantity of returns. Used product returns in remanufacturing may show extremely uncontrolled product condition variability, varying from minor to major damage and sometimes requiring repair in advance. In addition, the conditions of incoming cores are unknown to the remanufacturer in advance, and because of variations in usage patterns, they differ from one another. In order to identify quality levels of products and to justify the economic feasibility of remanufacturing operations, the recovery, inspection, and sorting of cores is worth implementing. In summary, quality uncertainty needs to be considered in the decision-making process of core acquisition.

Several factors are known to influence quality uncertainty during core acquisition in remanufacturing. First, the usage patterns and the end of life conditions of used products are not known or fully understood in detail, and this contributes to the increase in uncertainty. Many usage conditions, like maintenance history and operating conditions, are processes that the remanufacturer does not fully understand, and it would be expensive and time-consuming to obtain all the relevant data. Second, the measurement errors that arise between inspectors or from improper gauging are generally seen as a strong contributor to quality uncertainty.

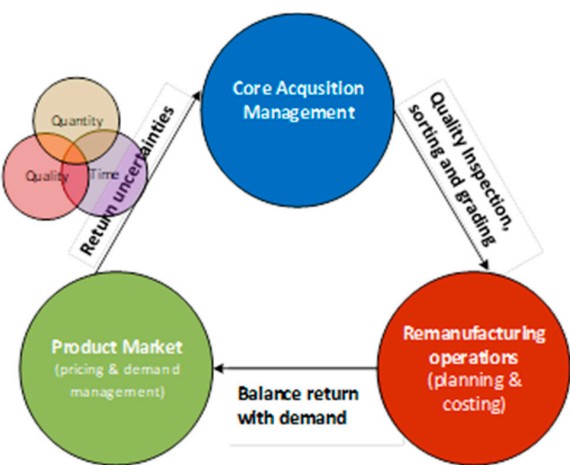

**Figure 1.** The uncertainties in remanufacturing (adapted and extended from Wei et al. [24].

There are a number of studies that have suggested an application of grey system theory to handle uncertainty in remanufacturing. These methods are effective in evaluation processes and for reducing uncertainty in decision-making processes regarding remanufacturing systems. Golinska et al. [18] proposed a tool for decision-making based on grey system theory to provide the classification of the sustainability level of remanufacturing. They categorized three classes of remanufacturers, which related to the different sustainability levels. This tool aids to define the existing state of remanufacturing and then recognize and prioritize company activities that need to be improved. Following their previous research, Golinska et al. [25] developed a mixed method for the sustainability assessment of remanufacturing processes by employing grey decision making for small- and medium-sized remanufacturers. The levels of sustainability were classified into three levels, namely, acceptable (requires minor improvement actions), conditionally acceptable (major corrective actions), and unacceptable (full corrective actions). Xin [19] evaluated the remanufacturability of used automotive components based on a grey fixed weight clustering approach. The results showed that the proposed indices could be applied for effective evaluation. Most studies in the field of grey system theory for handling uncertainty in remanufacturing have only focused on remanufacturing operations. The above literature review presents some gaps in the research on the application of grey system theory to handle quality uncertainty regarding sorting during core acquisition in remanufacturing, as detailed in Table 3. The differences between the scopes and proposed models from the past studies are evident in the table. One of the substantial differences between the proposed model and previous works is that in the present study, the evaluation criteria for incoming core quality are considered in the sorting model. In achieving the research objectives, this study contributes to filling the gap in the scope of core acquisition management and determining quality evaluation criteria for incoming cores by considering technological, physical, and usage conditions in remanufacturing in the context of a circular economy.

**Table 3.** Comparison of previous works with the proposed models. AHP: analytical hierarchy process.

| | | Research Contribution | | | | |
|---|---|---|---|---|---|---|
| | | Remanufacturing Activities | | Evaluating Criteria | | |
| References | Methods | Cores Acquisition | Remanufacturing Operations | Sustainability | Remanufacturability | Quality |
| Golinska et al. [18] | Grey clustering | | √ | 1. Economic 2. Environment 3. Social | | |

**Table 3.** *Cont.*

| References | Methods | Research Contribution | | | | |
|---|---|---|---|---|---|---|
| | | Remanufacturing Activities | | Evaluating Criteria | | |
| | | Cores Acquisition | Remanufacturing Operations | Sustainability | Remanufacturability | Quality |
| Golinska et al. [25] | Grey clustering | | √ | 1. Economic<br>2. Environment<br>3. Social | | |
| Xin [19] | Grey clustering | | √ | | 1. Economy<br>2. Technical<br>3. Resource<br>4. Energy and environ­ment | |
| Proposed model | AHP-entropy and grey clustering | √ | | | | 1. Technological condition<br>2. Physical condition<br>3. Usage condition |

## 3. Materials and Methods

An approach to deal with uncertain core quality during acquisition is the use of a multi-criteria decision-making model that employs an AHP-entropy approach in combination with grey clustering. The development of a quality sorting model was performed by combining AHP-Shannon entropy and grey clustering methods. These methods are particularly useful for studying multi-criteria quality evaluation for a set of incoming cores to be categorized into several classes with uncertain information. A descriptive flowchart depicting the major steps of the proposed methodology is depicted in Figure 2.

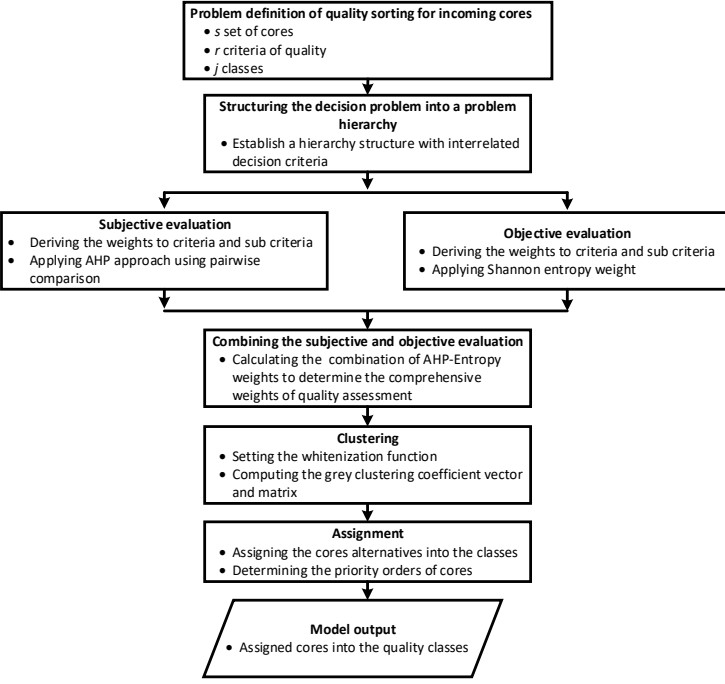

**Figure 2.** Major steps of the proposed methodology.

One of important parts of a multi-criteria quality sorting model is the weighting method. The broad use of the term "method" in multi-criteria decision-making is sometimes equated with a mathematical procedure to solve a problem by addressing and combining various routines and techniques [26]. Clustering weights could be obtained through different methods. Several methods currently exist for the measurement of weights. The pairwise comparison analysis facilitates a useful method to the handle relationship preferences for problems in subjective weighting. On the other hand, Shannon entropy weight is used as a measure of information uncertainty that is probabilistic in nature. Entropy basically means "loss", thus, the higher entropy, the lower content of the information, which represents unrecoverable information [26]. A major advantage of this method is that determining the weight allows an objective assessment. Later, in order to minimize the difference between subjective and objective weights, and additionally the deviation of the evaluation results [27], it is necessary to combine these weighting methods.

The development of a multi-criteria quality sorting model based on AHP-entropy and grey clustering for incoming cores in remanufacturing is illustrated in Figure 3. With the incoming core quality sorting problem, the key principle of a fixed grey clustering weight is used to consider the assessment data. These data were interpreted as object clusters, while the information collected was represented as distinct indicators.

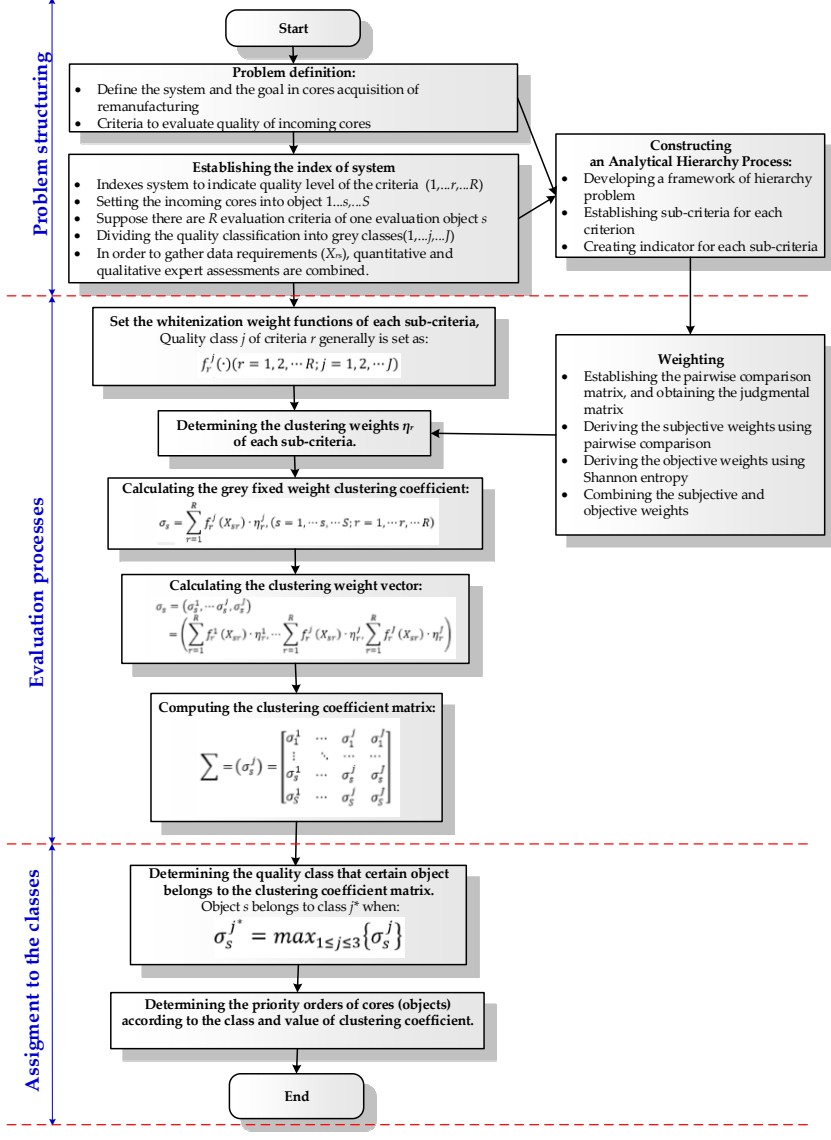

**Figure 3.** Flow diagram of the AHP grey clustering model for sorting.

There are several important data differences between quantitative and qualitative indicators. Quantitative data can be measured using a regular formula, whereas qualitative data need to be quantified by assessment instruments according to expert experience. Experts may include middle to senior professionals (managers or assistance managers) at a remanufacturing company with industrial experience not less than ten years. Experts may be from various departments, namely, production, quality control, and maintenance. For the collection of their subjective judgments based on a pair-wise questionnaire and concerns, a Likert scale should be used to answer such questions.

Generally speaking, the quality sorting model based on AHP-entropy grey clustering is divided into three stages, namely, problem structuring, evaluation, and class assignment. The method of clustering based on grey possibility functions is primarily used to verify whether or not the observational objects belong to pre-determined classes [17]. Overall, the decision model can be described as follows.

First, the structuring problem focuses on the development of effective structuring for the problem situation into a hierarchical structure of decisions, thereby defining the criteria to be evaluated for the goal when using AHP and entropy weighting. The purpose of problem structuring in the methodology is to help decision-makers better understand the problem, better explain and justify the conclusions, and finally ease the validation process. The development of an effective structure for the sorting problem hierarchy requires criteria to evaluate the incoming core quality in order to divide the cores into several classes and determine core alternatives. With this problem, "criteria" refers to the quality conditions of products, i.e., the technological, physical, and usage conditions. To classify the incoming core quality, $J$ quality classes are considered, and we refer to the quality class $j$ in subscript to differentiate between the different qualities of cores, i.e., $j = 1, 2, \ldots, J$. Moreover, classifying the $s^{th}$ core into the $j^{th}$ grey class based on the observed value $X_{sr}$ of the $s^{th}$ core judged against the $r^{th}$ criterion is referred to as grey clustering. The smaller $j$ is, the better the quality class of the core. The quality levels of the cores in the same quality class indicate the same remanufacturing process requirements.

Secondly, the evaluation process is a systematic process of collecting data, comparing preferences, and weighting to determine the clustering coefficient. Once the data are obtained, the next step is to assess the relative importance between criteria and sub-criteria at each level. For this reason, the AHP uses pairwise comparison judgement and entropy weighting to evaluate the strengths of importance. The following process can be carried out via adjustment of the whitenization weight functions. The classification of $S$ incoming cores into $J$ grey classes using the $r^{th}$ criterion is known as the $r^{th}$ criterion sub-classification by the whitenization weight function of the $r^{th}$ criterion $j^{th}$ sub-class referred as $f_r^j(\cdot)$. When the whitenization weight function $f_r^j(\cdot)$ takes the typical form as can be seen as Figure 4, then $X_r^j(1)$, $X_r^j(2)$, $X_r^j(3)$, $X_r^j(4)$ denotes the turning points of $f_r^j(\cdot)$. The second step in the evaluation process is determining the whitenization weight functions of each sub-criterion. Moreover, in the grey weight clustering process, the whitenization weight function can be grouped into subclass $j$ of criterion $r$. Then, the third step in the evaluation process is determining a clustering weight for each criterion $r$ ($\eta_r$). When assessing the quality levels of incoming cores, it is important to understand how important each criteria and sub-criteria are. In general, the weights between criteria have to be calculated by experts or inspectors. To obtain the weights, the AHP and Shannon entropy methods are used. Pairwise comparisons are generated using a nine-point scale with a standard AHP system that transforms expert preferences into available alternatives, such as equal, moderately, strongly, very strongly, or extremely preferred. Once the estimation of the relative preferences of the pairwise comparison matrix and entropy weights are obtained, it is necessary to compromise between them to determine the clustering weights of each

criterion. Following determination of the clustering weight, calculation of the grey fixed weight clustering coefficient ($\sigma_s$) is carried out as follows:

$$\sigma_s = \sum_{r=1}^{R} f_r^j(X_{sr}) \cdot \eta_r^j, \ (s = 1, \cdots s, \cdots S; r = 1, \cdots r, \cdots R) \tag{1}$$

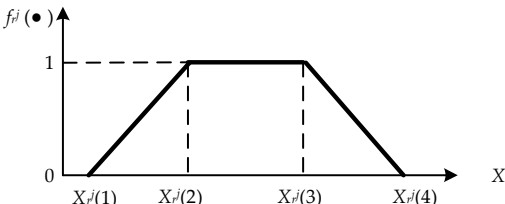

**Figure 4.** The typical $r^{th}$ criterion $j^{th}$ sub-class whitenization weight function.

Subsequently, the clustering weight vector is calculated by each class's weight coefficient ($\sigma_s^j$):

$$\begin{aligned}\sigma_s \ &= \ \left(\sigma_s^1, \cdots \sigma_s^j, \sigma_s^J\right) \\ &= \ \left(\sum_{r=1}^{R} f_r^1(X_{sr}) \cdot \eta_r^1, \cdots \sum_{r=1}^{R} f_r^j(X_{sr}) \cdot \eta_r^j, \sum_{r=1}^{R} f_r^J(X_{sr}) \cdot \eta_r^J\right)\end{aligned} \tag{2}$$

The results are transformed into a clustering coefficient matrix based on the calculation of the clustering weight vectors:

$$\sum = \left(\sigma_s^j\right) = \begin{bmatrix} \sigma_1^1 & \cdots & \sigma_1^j & \sigma_1^J \\ \vdots & \ddots & \cdots & \cdots \\ \sigma_s^1 & \cdots & \sigma_s^j & \sigma_s^J \\ \sigma_S^1 & \cdots & \sigma_S^j & \sigma_S^J \end{bmatrix} \tag{3}$$

The final step in this model is to assign the core (object) alternatives into classes by determining the class based on the clustering coefficient matrix. Therefore, core *s* belongs to class *j\** if:

$$\sigma_s^{j^*} A = max_{1 \leq j \leq 3}\left\{\sigma_s^j\right\} \tag{4}$$

The symbol of asterisk (*) is used to refer local maximum within clustering coefficient vector. Finally, we can determine the priority orders of cores based on the classes and values of the clustering coefficients.

## 4. Results and Discussion

In the heavy-duty equipment market, remanufacturing is likely to be associated with high-value parts in mechanical products. Many researchers [28–32] have utilized a case study of heavy-duty equipment remanufacturing to show the theoretical and practical contributions of their research. A case study was recently reported by Jiang et al. [32] that showed the feasibility of a proposed approach to achieve remanufacturing ecological efficiency assessment via a study of remanufacturing for hydraulic cylinders from remanufacturing enterprises in China.

In the case of Indonesia, the limited national supply of heavy-duty equipment manufacturers represents about 20% of the total demand for heavy equipment [33], where one of the problems is that the basic materials for machines and hydraulic parts in heavy equipment are still imported by the state. This has led companies to prefer to remanufacture the used parts to shorten lead times and increase part availability. The following emergent case study was identified from an Indonesian remanufacturer company for heavy-duty equipment parts. A used hydraulic cylinder is considered in this study to validate the

proposed model. The used hydraulic cylinder has a high added value and a strong opportunity for remanufacturing. In business practice, many remanufacturing companies have recovered used hydraulic cylinders for heavy-duty equipment parts (front suspension, rear suspension, and hoist cylinders) for mining and heavy-duty equipment construction.

### 4.1. Problem Structuring

Although eight dimensions of quality for new products have been successfully proposed by Garvin (1987), these dimensions cannot always be recognized when evaluating the quality levels of used products. This is due to the fact that when used products are at the end of their life phase, they cannot fulfill their main operating characteristics. Therefore, there remains a need for compatible quality criteria with used products. In order to provide quality criteria for used products in remanufacturing, Mustajib et al. [34] established that the quality criteria for sorting incoming cores can be assessed based on technological, physical, and usage conditions. The technological condition is a crucial criterion for assessing used products since it indicates the particular state of ability and complexity of technology that would influence the quality of end life product. Physical condition is also a very important criterion in the quality sorting of incoming cores because it is used to describe the body of the used product as can be seen and measured, i.e., it is a measure of physical damage. Meanwhile, the usage condition is a vital criterion for assessing the period of use and the maintenance that happens within a particular period. Moreover, these criteria can be used for quality evaluation of the remanufacturing of heavy-duty equipment parts, such as sorting for hydraulic cylinders. In successive steps, these criteria and sub-criteria may then be organized into a hierarchy descending from the overall goal or objective to the different stages and corresponding sub-criteria, as can be seen in Figure 5.

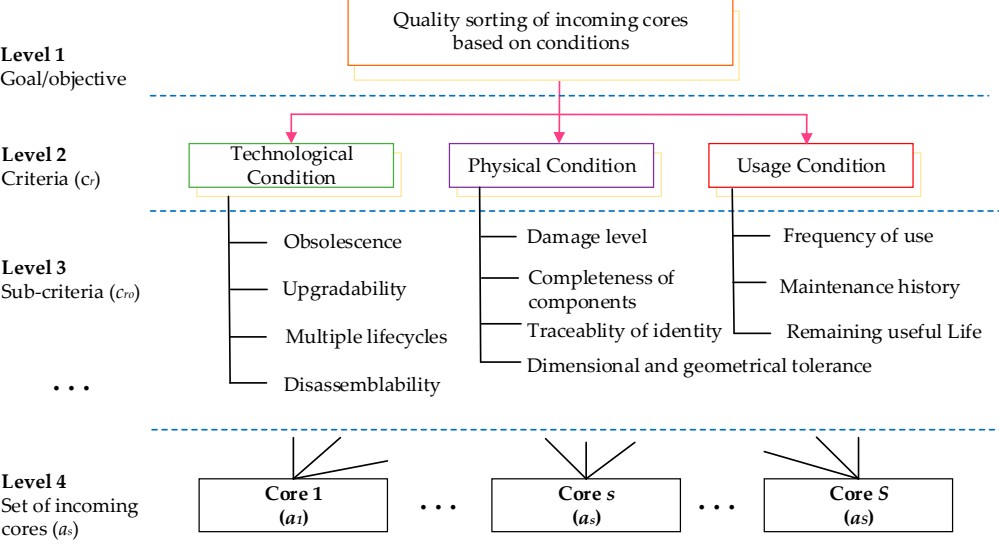

**Figure 5.** Hierarchy of decision levels for the incoming core quality sorting problem.

Sub-criteria were selected here as defined from previous works on the basis of both the literature review and practical considerations. The sub-criteria indicators can be calculated in a simplified way based on the expertise of decision-makers in the absence of detailed data for estimation, as seen in Tables A1 and A2. Furthermore, the sub-criteria for each criterion are defined as follows.

#### 4.1.1. Technological Condition
Obsolescence

Rapid innovation and technology development has led to a significant factor to shorten the life cycles of products and thus obsolescence. A product has technical or functional

obsolescence when customers are more interested in new products with better quality performance as a result of the introduction of new technology. To assess grades for technical obsolescence, Gao et al. [35] proposed five criteria to guide the qualitative evaluation of used products.

Upgradability

An upgrade is technical mitigation to handle the uncertain quality of used products. Meanwhile, the term "upgradeability" here refers to the potential level of used products to be upgraded efficiently and effectively to maintain an admissible presence on the market. Upgradeability represents the relative technological ease or viability of fostering continuous system renewal and enhancement at the level of engineering characteristics, parts, and the overall system. Remanufacturing with component upgrades may be an efficient alternative to used product obsolescence. In addition, an upgrade action in remanufacturing will improve the reliability of the used product. To evaluate the feasibility of upgrading, Du et al. [36] proposed an index that could be assessed by an accuracy improvement.

Multiple Life Cycles

The principle of multiple life cycles for products is a key technique in product development for remanufacturing. Since it is one of the strategies for prolonging a product life cycle after the end of life. Durable products are more effective for multiple life cycles. To assess the average life cycle number for a component, Geyer et al. [37] proposed a quantitative approach by dividing the average component life by the average product use.

Disassemblability

Disassembly is characterized as a complete assembly being dismantled down to its parts. Meanwhile, disassemblability can be loosely described as a level of ease with which a used product could be disassembled. The prerequisite of disassembly for remanufacturing is that it is non-destructive. Difficulty in obtaining parts may also contribute to a long time and high cost of disassembly. The principle of ease of disassemblability represents disassembly without force and via simple mechanisms [38]. For this reason, Gao et al. [35] and Xing et al. [39] used five criteria to provide a qualitative assessment of disassemblability. Meanwhile, Nof et al. [40] recommended a guideline for easy disassembly. On the contrary, Du et al. [36] quantified the index of ease of disassembly. Similarly, Ali [41] has suggested comprehensive methods to quantitatively evaluate disassemblability based on product design, process technology, and incoming quality assessment.

4.1.2. Physical Condition

Damage Level

Used products may have degraded features with different degrees of damage. This indicator is typically calculated by fault features such as corrosions, cracks, wear, and so on. The damage level can be quantified and classified according to the degree of damage [32,33,42,43].

Component Completeness

Completeness is defined by Yoe [44] as all the necessary parts being accounted for and included. It means that the incoming core should not be broken down into constituent parts and should be delivered entirely without missing parts. In case of any uncertainty as to the completeness of the incoming core, an extended inspection is carried out to verify the inner structure of the used product [25].

Traceability of Identity

Product identification and traceability are essential for the quality acceptance of used products. The core should have original equipment such as the manufacturer's identification number (e.g., manufacturer stickers) because of the wide variety of products,

allowing the model, type, and parameters to be recognized [25]. The availability of a used product's identity, such as text, readable labels, and barcodes that are not missing or fading over the use phase of a product's life, allows easy recognition for quality sorting.

Dimensional and Geometrical Tolerance

The term dimensional tolerance is generally understood as the minimum and maximum values allowable for the dimension of the parts to work properly. Meanwhile, geometric tolerance is a significant assessment factor that depends on the consistency of the body shapes of used products. When classified, the sorted parts of used products will be reprocessed based on the quality loss degree according to an allowable tolerance. If the used products have the highest deformation from ideal dimension and shape (exceeding the acceptable tolerance), they cannot be reconditioned to their original performance by remanufacturing and they are then recycled [45].

### 4.1.3. Usage Condition
Frequency of Use

The usage condition is influenced by user behavior during the use phase of a product. During the middle of a product's life, a product's performance deteriorates with the frequency of use and operating conditions. This is evident in the case of the quality of used parts in heavy equipment that is evaluated based on hours of operation.

Remaining Useful Life

The term remaining useful life (RUL) has been used to refer to prediction for determining whether used parts can be reused. When used for a while, the remaining life of a used part can be defined as within a residual operational lifetime that can be predicted. To assess the degree of the remaining life of the used product, products can be categorized into several levels according to the minimum and maximum values of the obtained remaining useful life values [32,46]

Maintenance History

Maintenance characterizes a set of actions taken to allow a product to work at a predetermined level during the use phase. A proper maintenance strategy will extend the original life of a product and make it compatible with remanufacturing [47]. Due to the potentially lower reliability of used products, an appropriate maintenance strategy is required. Moreover, Stadnicka et al. [48] have proposed criteria to carry out the classification of equipment based on the failure frequency (amount of failure registry entries each year).

### 4.2. Evaluation Processes

To measure subjective weights, experts were asked to estimate relative preferences for criteria and sub-criteria based on their expert knowledge. Table 4 provides pairwise comparisons among the three core quality condition criteria. The estimations for the relative preferences for sub-criteria are presented in Tables 5–7 as follows. Thereafter, based on the previous data from the pairwise comparison matrix, then get global weights for each sub-criterion are shown in in Tables 8–10. In this study, four experts were selected to serve as evaluators in order to estimate subjective weights. The criteria for selecting the experts were the following: middle to senior professionals of a heavy-duty manufacturing company, with industrial experience not less than ten years in a respective field of production, quality, or maintenance.

**Table 4.** Matrix for pairwise comparisons for the main criteria.

|  | $c_1$ | $c_2$ | $c_3$ |
|---|---|---|---|
| $c_1$ | 1 | 5 | 3 |
| $c_2$ | 1/5 | 1 | 3 |
| $c_3$ | 1/3 | 1/3 | 1 |

**Table 5.** Matrix for pairwise comparison of the sub-criteria of the technological conditions.

|  | $c_{11}$ | $c_{12}$ | $c_{13}$ | $c_{14}$ |
|---|---|---|---|---|
| $c_{11}$ | 1 | 1/5 | 1/9 | 3 |
| $c_{12}$ | 5 | 1 | 1/7 | 1/7 |
| $c_{13}$ | 9 | 7 | 1 | 9 |
| $c_{14}$ | 1/3 | 7 | 1/9 | 1 |

**Table 6.** Matrix for pairwise comparison of the sub-criteria of the physical condition.

|  | $c_{21}$ | $c_{22}$ | $c_{23}$ | $c_{24}$ |
|---|---|---|---|---|
| $c_{21}$ | 1 | 1/3 | 5 | 5 |
| $c_{22}$ | 3 | 1 | 1/3 | 5 |
| $c_{23}$ | 1/5 | 3 | 1 | 5 |
| $c_{24}$ | 1/5 | 1/5 | 1/5 | 1 |

**Table 7.** Matrix for pairwise comparison of the sub-criteria of the usage condition.

|  | $c_{31}$ | $c_{32}$ | $c_{33}$ |
|---|---|---|---|
| $c_{31}$ | 1 | 1 | 3 |
| $c_{32}$ | 1 | 1 | 7 |
| $c_{33}$ | 1/3 | 1/7 | 1 |

**Table 8.** Weights of sub-criteria related to the technological condition.

| | Technological Condition, $c_1$ (Weight 0.62) | | | |
|---|---|---|---|---|
| Sub-criteria | $c_{11}$ | $c_{12}$ | $c_{13}$ | $c_{14}$ |
| Local weights | 0.10 | 0.13 | 0.62 | 0.16 |
| Global weights | 0.062 | 0.080 | 0.384 | 0.099 |

**Table 9.** Weights of sub-criteria related to the physical condition.

| | Physical Condition, $c_2$ (Weight 0.24) | | | |
|---|---|---|---|---|
| Sub-criteria | $c_{21}$ | $c_{22}$ | $c_{23}$ | $c_{24}$ |
| Local weights | 0.34 | 0.32 | 0.29 | 0.05 |
| Global weights | 0.082 | 0.077 | 0.070 | 0.012 |

**Table 10.** Weights of sub-criteria related to the usage condition.

| | Usage Condition, $c_3$ (Weight 0.14) | | |
|---|---|---|---|
| Sub-criteria | $c_{31}$ | $c_{32}$ | $c_{33}$ |
| Local weights | 0.39 | 0.51 | 0.10 |
| Global weights | 0.055 | 0.072 | 0.014 |

Nine ($S = 9$) incoming cores of used hydraulic cylinders were acquired by a remanufacturer and needed to be classified according to the criteria which were then expanded into 11 sub-criteria ($R = 11$). One common classification for used parts in heavy-duty equipment remanufacturing is to have three quality classes, such as in Toromon Cat [49].

Moreover, the used hydraulic cylinders were sorted into three distinctive grey classes ($J = 3$), namely, namely, either best, middle, or worst quality. In fact, these classes were associated with three groups of core acceptance criteria for used hydraulic cylinders based

on visual/mechanical inspection, along with a full core refund for the best quality class, partial core refund for the middle quality class, and no core refund for the worst quality class. These classes are in line with the recommendations of previous research. For example, Thierry et al. [50] suggested three different quality categories for product reuse, recovery, and waste management.

The classification for the $s^{th}$ core into the $j^{th}$ grey class according to the observed value of the $s^{th}$ core judged against the $r^{th}$ criterion is denoted by $X_{sr}$. Due to the high uncertainty in the core conditions, sometimes it is very difficult to determine the technical index quantitatively for each criterion due to the complexity and difficulty. Therefore, it can only be measured qualitatively by expert assessment as can be seen in Table A1. The assessment value of the incoming core quality condition data is shown in Table 11.

**Table 11.** The observed values of each criterion for each incoming core ($X_{sr}$).

| Incoming Core Set ($a_s$) | Sub-Criteria ($c_{ro}$) and the Global Weights ($\eta$) | | | | | | | | | | |
|---|---|---|---|---|---|---|---|---|---|---|---|
| | $c_{11}$ | $c_{12}$ | $c_{13}$ | $c_{14}$ | $c_{21}$ | $c_{22}$ | $c_{23}$ | $c_{24}$ | $c_{31}$ | $c_{32}$ | $c_{33}$ |
| | 0.062 | 0.080 | 0.384 | 0.099 | 0.082 | 0.077 | 0.070 | 0.012 | 0.055 | 0.072 | 0.014 |
| $a_1$ | 3 | 3 | 4 | 2 | 2 | 0.8 | 3 | 2 | 3 | 4 | 4 |
| $a_2$ | 4 | 3 | 3 | 3 | 1 | 0.9 | 4 | 2 | 2 | 3 | 3 |
| $a_3$ | 2 | 4 | 5 | 2 | 2 | 0.9 | 4 | 1 | 2 | 4 | 4 |
| $a_4$ | 3 | 3 | 5 | 2 | 2 | 0.8 | 5 | 3 | 3 | 3 | 5 |
| $a_5$ | 3 | 4 | 4 | 2 | 2 | 0.9 | 3 | 2 | 1 | 4 | 4 |
| $a_6$ | 4 | 4 | 3 | 3 | 3 | 0.85 | 4 | 1 | 2 | 3 | 3 |
| $a_7$ | 4 | 2 | 4 | 2 | 2 | 0.88 | 4 | 2 | 2 | 4 | 4 |
| $a_8$ | 5 | 3 | 5 | 3 | 1 | 0.9 | 3 | 3 | 2 | 3 | 5 |
| $a_9$ | 2 | 4 | 5 | 2 | 3 | 0.95 | 3 | 2 | 3 | 3 | 4 |

Furthermore, to assess the qualitative indicators, the expert assessments shown in Table A2 were used. Thus, the values for the upper, middle, and lower classes could be obtained by applying the whitenization weight function (Figure 6) as proposed by Formula (5) to (13) as follows.

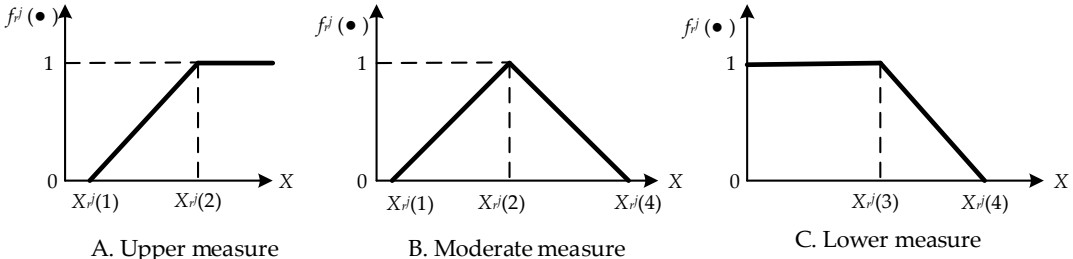

**Figure 6.** Typical whitenization function for the quality sorting problem.

For example, criterion $c_{22}$ with the whitenization weight functions is given as:

$$f_r^1(X_{rs}) = \begin{cases} 0, & X < 0\% \\ \frac{X}{80}, & 0\% \leq X < 80\% \\ 1, & 80\% \leq X \leq 100\% \end{cases} \quad (5)$$

$$f_r^2(X_{rs}) = \begin{cases} 0, & X < 0 \\ \frac{X}{75}, & 0 \leq X \leq 75\% \\ \frac{100-X}{25}, & 75\% < X \leq 100\% \\ 0, & X > 100\% \end{cases} \quad (6)$$

$$f_r^3(X_{rs}) = \begin{cases} 0, & X < 0 \\ \frac{100-X}{100}, & 0\% \le X \le 100\% \\ 0, & X > 100 \end{cases} \tag{7}$$

Meanwhile, for the criteria with $r$ $c_{11}$, $c_{12}$, $c_{13}$, $c_{14}$, $c_{23}$, $c_{32}$, $c_{33}$, the whitenization weight functions are expressed as:

$$f_r^1(X_{rs}) = \begin{cases} 0, & X < 0 \\ \frac{X}{5}, & 0 \le X < 5 \\ 1, & X \ge 5 \end{cases} \tag{8}$$

$$f_r^2(X_{rs}) = \begin{cases} 0, & X < 0 \\ \frac{X}{2.5}, & 0 \le X \le 2.5 \\ \frac{5-X}{2.5}, & 2.5 < X \le 5 \\ 0, & X > 5 \end{cases} \tag{9}$$

$$f_r^3(X_{rs}) = \begin{cases} 0, & X < 0 \\ 1, & 0 \le X \le 2.5 \\ \frac{5-X}{2.5}, & 2.5 < X \le 5 \\ 0, & X > 5 \end{cases} \tag{10}$$

Moreover, for the criteria with $r$ $c_{21}$, $c_{24}$ $c_{31}$, the whitenization weight functions are expressed as:

$$f_r^1(X_{rs}) = \begin{cases} 0, & X < 0 \\ \frac{5-X}{5}, & 0 \le X < 5 \\ 1, & X \ge 5 \end{cases} \tag{11}$$

$$f_r^2(X_{rs}) = \begin{cases} 0, & X < 0 \\ \frac{X}{2.5}, & 0 \le X \le 2.5 \\ \frac{5-X}{2.5}, & 2.5 < X \le 5 \\ 0, & X > 5 \end{cases} \tag{12}$$

$$f_r^3(X_{rs}) = \begin{cases} 0, & X < 0 \\ 1, & 0 \le X \le 2.5 \\ \frac{X}{5}, & 2.5 \le X \le 5 \\ 0, & X > 5 \end{cases} \tag{13}$$

### 4.3. Assignment to the Classes

Based on the whitenization weight function $f_r^j(\cdot)$ formulated in Equations (5)–(13) and the clustering global weights $\eta$, the grey fixed weight cluster coefficients for each class $(\sigma_s^j)$ can be found. The calculation of the grey fixed weight clustering coefficients as decision indicators is summarized in Table 12. What stands out in Table 12 is that cores $a_1, a_3, a_4, a_5, a_6, a_8, a_9$ were mostly classified into class 1. Meanwhile, cores $a_2$, $a_7$ were essentially classified into class 2. Noticeably, no core was classified to class 3. These results indicate that the majority of the used hydraulic cylinders were of considerable quality.

**Table 12.** The value of the grey fixed weight cluster coefficient for each class $\left(\sigma_s^j\right)$ with subjective weighting.

| $a_s$ | $\sigma_s^j = \sum\limits_{r=1}^{R} f_r^j(X_r)\cdot\eta_r$ | | | Maximum Coefficient Value | Grey Class |
|---|---|---|---|---|---|
| | $j = 1$ | $j = 2$ | $j = 3$ | $\sigma_s^{j^*} = max_{1\leq j\leq 3}\{\sigma_s^j\}$ | |
| $a_1$ | **0.7092** | 0.5945 | 0.6804 | 0.7092 | $j^* = 1$ |
| $a_2$ | 0.6778 | **0.7249** | 0.6399 | 0.7249 | $j^* = 2$ |
| $a_3$ | **0.7888** | 0.4601 | 0.4443 | 0.7888 | $j^* = 1$ |
| $a_4$ | **0.789** | 0.4862 | 0.4078 | h0.789 | $j^* = 1$ |
| $a_5$ | **0.7362** | 0.5405 | 0.5071 | 0.7362 | $j^* = 1$ |
| $a_6$ | **0.6798** | 0.6757 | 0.65815 | 0.6798 | $j^* = 1$ |
| $a_7$ | 0.598 | **0.671** | 0.57804 | 0.671 | $j^* = 2$ |
| $a_8$ | **0.833** | 0.4037 | 0.57412 | 0.833 | $j^* = 1$ |
| $a_9$ | **0.7478** | 0.42325 | 0.43905 | 0.7478 | $j^* = 1$ |

The clustering weights also could be achieved through the objective weighting method, i.e., the Shannon entropy method. To establish the clustering weights for each criterion $r$ ($\eta_r$), the Shannon entropy weights ($w_{ro}^e$) were used. The Shannon entropy weights are summarized by the three steps given as follows [26,27]:

Step 1. Normalization

At the beginning, due to the data having different scales and units, the data should first be normalized into dimensionless data as follows:

$$p_{sr} = \frac{x_{sr}}{\sum_{s=1}^{S} x_{sr}} \tag{14}$$

The results are presented in Table 13.

**Table 13.** The normalized values of each criterion ($p_{sr}$) for each incoming core.

| Core ($a_s$) | Sub-Criteria ($c_{ro}$) | | | | | | | | | | |
|---|---|---|---|---|---|---|---|---|---|---|---|
| | $c_{11}$ | $c_{12}$ | $c_{13}$ | $c_{14}$ | $c_{21}$ | $c_{22}$ | $c_{23}$ | $c_{24}$ | $c_{31}$ | $c_{32}$ | $c_{33}$ |
| $a_1$ | 0.100 | 0.100 | 0.133 | 0.067 | 0.067 | 0.027 | 0.100 | 0.067 | 0.100 | 0.133 | 0.133 |
| $a_2$ | 0.133 | 0.100 | 0.079 | 0.111 | 0.100 | 0.097 | 0.083 | 0.067 | 0.067 | 0.100 | 0.100 |
| $a_3$ | 0.067 | 0.133 | 0.132 | 0.056 | 0.100 | 0.129 | 0.111 | 0.033 | 0.067 | 0.133 | 0.133 |
| $a_4$ | 0.100 | 0.100 | 0.132 | 0.167 | 0.150 | 0.097 | 0.139 | 0.100 | 0.100 | 0.100 | 0.167 |
| $a_5$ | 0.100 | 0.133 | 0.105 | 0.111 | 0.050 | 0.129 | 0.111 | 0.067 | 0.033 | 0.133 | 0.133 |
| $a_6$ | 0.167 | 0.133 | 0.079 | 0.056 | 0.100 | 0.097 | 0.083 | 0.033 | 0.067 | 0.100 | 0.100 |
| $a_7$ | 0.133 | 0.067 | 0.105 | 0.111 | 0.100 | 0.129 | 0.111 | 0.067 | 0.067 | 0.133 | 0.133 |
| $a_8$ | 0.167 | 0.100 | 0.132 | 0.167 | 0.100 | 0.097 | 0.139 | 0.100 | 0.067 | 0.100 | 0.167 |
| $a_9$ | 0.067 | 0.133 | 0.132 | 0.111 | 0.150 | 0.097 | 0.111 | 0.067 | 0.100 | 0.100 | 0.133 |

Step 2. Calculation of the entropy ($E_r$) and the degree of diversity ($d_r$)

Afterwards, entropy within the data of the normalized decision matrix for the $r^{th}$ criterion can be obtained using Equation (15). Meanwhile, the results are given in Table 14.

$$E_r = -c \sum_{s=1}^{S} p_{sr} \ln(p_{sr}) \tag{15}$$

where the constant $c$ is defined as:

$$c = \frac{1}{\ln(S)} \tag{16}$$

Later on, in order to calculate the amount of uncertainty, it can be stated by the degree of divergence ($d_r$), which is calculated as follows:

$$d_r = 1 - E_r \tag{17}$$

**Table 14.** The entropy values ($E_r$) of each criterion for each incoming core.

| Core ($a_s$) | Sub-Criteria ($c_{ro}$) | | | | | | | | | | |
|---|---|---|---|---|---|---|---|---|---|---|---|
| | $c_{11}$ | $c_{12}$ | $c_{13}$ | $c_{14}$ | $c_{21}$ | $c_{22}$ | $c_{23}$ | $c_{24}$ | $c_{31}$ | $c_{32}$ | $c_{33}$ |
| $a_1$ | −0.100 | −0.100 | −0.117 | −0.078 | −0.078 | −0.042 | −0.100 | −0.078 | −0.100 | −0.117 | −0.117 |
| $a_2$ | −0.117 | −0.100 | −0.087 | −0.106 | −0.100 | −0.098 | −0.090 | −0.078 | −0.078 | −0.100 | −0.100 |
| $a_3$ | −0.078 | −0.117 | −0.116 | −0.070 | −0.100 | −0.115 | −0.106 | −0.049 | −0.078 | −0.117 | −0.117 |
| $a_4$ | −0.100 | −0.100 | −0.116 | −0.130 | −0.124 | −0.098 | −0.119 | −0.100 | −0.100 | −0.100 | −0.130 |
| $a_5$ | −0.100 | −0.117 | −0.103 | −0.106 | −0.065 | −0.115 | −0.106 | −0.078 | −0.049 | −0.117 | −0.117 |
| $a_6$ | −0.130 | −0.117 | −0.087 | −0.070 | −0.100 | −0.098 | −0.090 | −0.049 | −0.078 | −0.100 | −0.100 |
| $a_7$ | −0.117 | −0.078 | −0.103 | −0.106 | −0.100 | −0.115 | −0.106 | −0.078 | −0.078 | −0.117 | −0.117 |
| $a_8$ | −0.130 | −0.100 | −0.116 | −0.130 | −0.100 | −0.098 | −0.119 | −0.100 | −0.078 | −0.100 | −0.130 |
| $a_9$ | −0.078 | −0.117 | −0.116 | −0.106 | −0.124 | −0.098 | −0.106 | −0.078 | −0.100 | −0.100 | −0.117 |
| $E_r$ | **0.777** | **0.763** | **0.763** | **0.698** | **0.699** | **0.713** | **0.751** | **0.537** | **0.590** | **0.803** | **0.835** |

Step 3. Calculation of objective weights ($w_{ro}^e$)

The last step is the linear normalization of $d_r$ to find the relative weight of each criterion is calculated using Equation (18). The results are presented in Table 15.

$$w_{ro}^e = \frac{d_r}{\sum_{r=1}^{R} d_r} \qquad (18)$$

**Table 15.** The divergence ($d_r$) and entropy weights ($w_{ro}^e$) of each criterion for each incoming core.

| | Sub-Criteria ($c_{ro}$) | | | | | | | | | | |
|---|---|---|---|---|---|---|---|---|---|---|---|
| | $c_{11}$ | $c_{12}$ | $c_{13}$ | $c_{14}$ | $c_{21}$ | $c_{22}$ | $c_{23}$ | $c_{24}$ | $c_{31}$ | $c_{32}$ | $c_{33}$ |
| Divergence | 0.223 | 0.237 | 0.237 | 0.302 | 0.301 | 0.287 | 0.249 | 0.463 | 0.410 | 0.197 | 0.165 |
| Entropy weights | 0.073 | 0.077 | 0.077 | 0.098 | 0.098 | 0.093 | 0.081 | 0.151 | 0.134 | 0.064 | 0.054 |

Finally, according to the AHP global weights $\left(w_{ro}^h\right)$ which are equal to $\eta_{ro}$ in Table 11 and the entropy weights ($w_{ro}^e$) in Equation (18), the weight of each criterion can be acquired on the principle of combined weighting methods as established by Equation (15). The weights of each criterion were derived by the combined weighting methods ($w_{ro}^c$) given in Table 16.

$$w_{ro}^c = \frac{w_{ro}^h \times w_{ro}^e}{\sum_{r=1}^{R} w_{ro}^h \times w_{ro}^e} \qquad (19)$$

**Table 16.** The combined weighting methods ($w_{ro}^c$).

| Methods | Sub-Criteria ($c_{ro}$) | | | | | | | | | | |
|---|---|---|---|---|---|---|---|---|---|---|---|
| | $c_{11}$ | $c_{12}$ | $c_{13}$ | $c_{14}$ | $c_{21}$ | $c_{22}$ | $c_{23}$ | $c_{24}$ | $c_{31}$ | $c_{32}$ | $c_{33}$ |
| AHP weights | 0.062 | 0.080 | **0.384** | **0.099** | **0.082** | 0.077 | 0.070 | 0.012 | 0.055 | 0.072 | 0.014 |
| Entropy weights | 0.073 | 0.077 | 0.077 | 0.098 | 0.098 | 0.093 | 0.081 | 0.151 | 0.134 | 0.064 | 0.054 |
| Combined weighting methods ($w_{ro}^c$) | 0.06 | 0.08 | **0.39** | **0.13** | 0.11 | 0.09 | 0.07 | 0.02 | 0.10 | 0.06 | 0.01 |

Figure 7 shows a clustered bar chart that was used to compare the weighting values between the pairwise comparison judgment, Shannon entropy, and combined weighting methods. As can be seen, the weighting methods gave different results for all three methods. What is interesting in the chart is the differences between weighted results between methods. It is clear from the graph that the sub-criterion of multiple life cycles is the most important factor in evaluating the quality of incoming cores based on subjective weighting. It is respectively followed by disassemblability as the second highest, and damage level as third highest. In contrast, according to objective weighting, it is clear from the graph that the sub-criterion of geometric and dimensional tolerance is more important than the multiple life cycle sub-criterion. Subsequently, sub-criterion of the geometric

and dimensional tolerance is slightly more important than sub-criterion of the frequency of use. Interestingly, there were also slightly different subjective weights in the results between the combined methods; however, multiple life cycles, disassemblability, and the damage level were still the three most important sub-criteria for the combined weighting methods. Altogether, we found that the values of the subjective weights more fluctuated than objective weights.

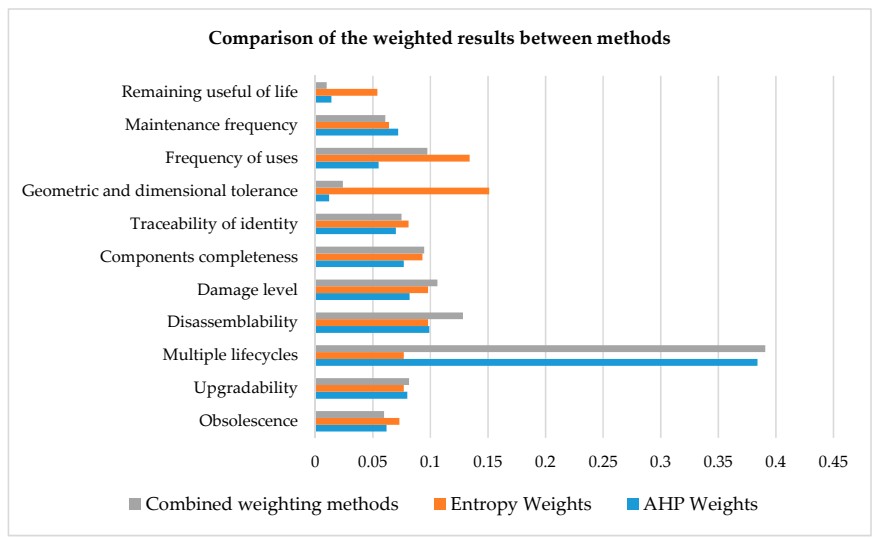

**Figure 7.** Comparison of the weighted results of the methods.

The overall values of the grey fixed weight cluster coefficients based on the Shannon entropy weight method are reported in Table 17. From this table, we can see that the clustering weights using the Shannon entropy method resulted in the highest of coefficient values for cores a5, a6, a8 in the best class; a2, a4, a9 into the middle class; and a1, a3, a7 into the worst class. These results indicate that the use of the Shannon entropy weight and grey clustering methods presumably acquired greater balance in the assignment process into overall quality classes. Nevertheless, the maximum grey clustering coefficient values based on combined weights were distinct between subjective and objective weighting. Therefore, these may have contributed to the changes in core assignment within classes as can be seen in Table 18. It is listed in the table that the number of the assigned cores within the worst class was minor.

**Table 17.** Grey fixed weight cluster coefficients for each class $\left(\sigma_s^j\right)$ based on entropy weight.

| $a_s$ | $\sigma_s^j = \sum\limits_{r=1}^{R} f_r^j(X_r)\cdot\eta_r$ | | | Maximum Coefficient Value | Grey Class |
|---|---|---|---|---|---|
| | $j = 1$ | $j = 2$ | $j = 3$ | $\sigma_s^{j^*} = max_{1\leq j\leq 3}\{\sigma_s^j\}$ | |
| $a_1$ | 0.6503 | 0.6184 | **0.8032** | 0.8032 | $j^* = 3$ |
| $a_2$ | 0.6877 | **0.7170** | 0.6590 | 0.7170 | $j^* = 2$ |
| $a_3$ | 0.6957 | 0.6368 | **0.7076** | 0.7076 | $j^* = 3$ |
| $a_4$ | 0.6455 | **0.6885** | 0.5569 | 0.6885 | $j^* = 2$ |
| $a_5$ | **0.6988** | 0.6097 | 0.6790 | 0.6988 | $j^* = 1$ |
| $a_6$ | **0.7172** | 0.6571 | 0.6249 | 0.7172 | $j^* = 1$ |
| $a_7$ | 0.6719 | 0.6419 | **0.7087** | 0.7087 | $j^* = 3$ |
| $a_8$ | **0.7082** | 0.6233 | 0.6262 | 0.7082 | $j^* = 1$ |
| $a_9$ | 0.6053 | **0.6950** | 0.6698 | 0.6950 | $j^* = 2$ |

**Table 18.** Grey fixed weight cluster coefficients for each class $\left(\sigma_s^j\right)$ based on combined weights.

| $a_s$ | $\sigma_s^j = \sum\limits_{r=1}^{R} f_r^j(X_r) \cdot \eta_r$ | | | Maximum Coefficient Value | Grey Class |
|---|---|---|---|---|---|
| | $j = 1$ | $j = 2$ | $j = 3$ | $\sigma_s^{j^*} = max_{1 \leq j \leq 3}\{\sigma_s^j\}$ | |
| $a_1$ | 0.6672 | 0.6014 | **0.7040** | 0.7040 | $j^* = 3$ |
| $a_2$ | 0.6600 | **0.7350** | 0.6850 | 0.7350 | $j^* = 2$ |
| $a_3$ | **0.7400** | 0.5150 | 0.5490 | 0.7400 | $j^* = 1$ |
| $a_4$ | **0.7260** | 0.5300 | 0.4780 | 0.7260 | $j^* = 1$ |
| $a_5$ | **0.7020** | 0.5590 | 0.6530 | 0.7020 | $j^* = 1$ |
| $a_6$ | 0.6740 | **0.6990** | 0.6965 | 0.6990 | $j^* = 2$ |
| $a_7$ | 0.5760 | **0.6820** | 0.6556 | 0.6820 | $j^* = 2$ |
| $a_8$ | **0.7900** | 0.4670 | 0.6460 | 0.7900 | $j^* = 1$ |
| $a_9$ | **0.7620** | 0.4915 | 0.4735 | 0.7620 | $j^* = 2$ |

As mentioned previously, this work aimed to propose a multi-criteria quality sorting model based on an AHP and entropy weights coupled with grey clustering to handle quality uncertainty for incoming cores. The clustering weights of the model could be acquired through subjective and objective methods. By comparing subjective and objective weighting, it was obvious that the output of the model produced significantly different results for the majority of cores assigned into pre-determined classes. It is evident from the results that the model with subjective weights gave significant results when classifying the qualities of incoming cores into two classes. By contrast, the model with objective weights gave significant results when classifying the qualities of incoming cores into three classes. We found that weighting methods may have a great influence on the assignment of classified cores. Figure 8 reveals a clustered column chart that was used to compare the results and the proportion of assigned cores across the quality classes when using the three weighting methods. What is striking in this chart is the phenomenon arising from the use of the subjective AHP pairwise comparison weighting method in terms of the proposed model, which gives a result where core assignment results tend to be dominant in certain classes. The use of objective entropy weighting produced results that were more balanced in terms of assigning cores into overall classes. Comparing AHP weights and entropy weights, it is obvious that a significant improvement in the use of combined methods in the proposed model could reduce possible weighting biases.

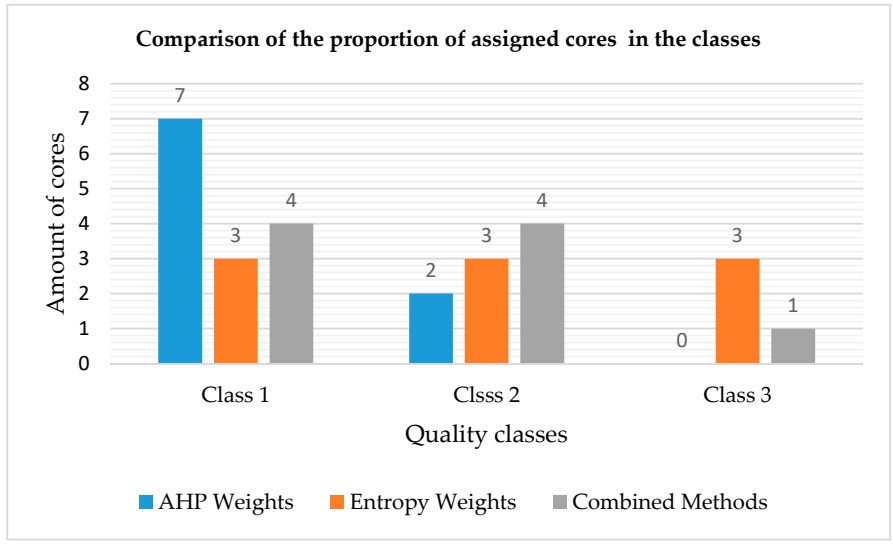

**Figure 8.** Impact of the weighting methods on the core assignment results.

Further results showed that this research has been successfully conducted in order to assess the importance of the quality levels of used products for heavy-duty equipment in terms of considering them with grey classes. One interesting finding is that assessment formulae can be obtained both qualitatively and quantitatively. Another important finding is that the grey quality classes of incoming cores can be found by combining the AHP-entropy weights and a grey clustering model. In summary, these results show the effectiveness of the proposed model to handle uncertainty in quality sorting problems for incoming cores in remanufacturing systems. The model considers limited data and is a novel sorting model for solving quality uncertainty problems in core acquisition management.

The findings of this work have some managerial implications. First, these findings have significant practical implications for the management of remanufacturing companies regarding the understanding of how to manage uncertain quality. The evidence from this study suggests that sorting problems should be solved by applying AHP-entropy weights and grey clustering as a short-term quality control, rather than a probability statistics approach as a solution for incomplete data obstacles. Second, this study has revealed that, in general, physical, technical, and usage conditions are important criteria for sorting of incoming core quality. In other words, visual inspection and sorting can be performed simultaneously based on the physical condition. Another implication of the current study is that it supports strong recommendations to focus on multiple-criteria decision-making for the quality sorting of incoming cores based on the expert knowledge. Third, the results seem to indicate that the combination of weights provides some support for the conceptual premise that possible biases in weighting will be reduced.

In summary, these results show that the proposed model could be applied without difficulty in order to handle uncertain core quality with limited data. This study strengthens the idea that sorting incoming cores into three classes in heavy-duty remanufacturing is a practical option. This is certainly true in the case of the three quality levels for incoming cores of reuse, repair, and scrapping.

Like other works, this research has its limitations. For example, the scope of this study was limited in terms of quality. The model could be extended by applying sustainability and remanufacturability as evaluation criteria. In addition, this model considers one-way relationships between criteria and sub-criteria. Considerably more work will need to be done to develop the sorting model with two-way relationships between criteria and sub-criteria.

## 5. Conclusions

Remanufacturing is a vital aspect in a circular economy as an effective means of achieving sustainable industry. The important challenge in remanufacturing is handling quality uncertainty with incoming cores. Determining how to reduce quality uncertainty is a crucial issue in remanufacturing systems. When reviewing the literature, no multi-criteria model was found to solve the quality sorting problem for core acquisition management. A quality sorting process can be viewed as a system composed of objectives, alternatives, criteria, resources, and assignment, and consequently each part is usually interrelated. This work was undertaken to propose a new hybrid approach (AHP-entropy and grey Clustering), whereas combinations of subjective and objective approaches have been used for weighting coupled with grey clustering using possibility functions to evaluate the quality conditions of used products as a principal approach for sorting problems to handle quality uncertainty with incoming cores. The quality criteria for sorting the incoming cores were considered according to the technological, physical, and usage conditions.

In general, the quality sorting model based on AHP-entropy grey clustering is divided into three stages, namely, problem structuring, evaluation, and class assignment. Problem structuring focuses on better understanding the quality sorting problem by developing an effective structure for the problem situation into a hierarchical structure of decisions. Thereafter, the evaluation focuses on collecting data, comparing preferences, and weighting in order to determine grey clustering coefficients. The clustering weights of the model

could be acquired through subjective and objective methods. The most obvious finding to emerge from this study is that the weights with subjective weighting tend to have more variance than with objective weighting. Combining these methods minimized the difference between subjective and objective weights, as well as the deviation of the evaluation results. The final step in this model was to assign the incoming cores into predetermined classes based on the maximum values of the grey clustering coefficients. The research has shown that multiple life cycles, disassemblability, and damage level are the three most important sub-criteria within the combined weighting methods. Overall, these results indicate that the combination of weights provides some support for the conceptual premise that possible biases in weighting will be reduced as a result of such practice. The findings here suggest that weighting methods during an evaluation process can have an effect on core assignment. To this respect, the results show how variations in sub-criteria weights affect cores assignment between quality classes. These results provide important insights for decision-makers considering hybrid approaches for quality criteria in sorting incoming cores.

The study was successful as it was able to develop a hybrid approach for the multi-criteria sorting of cores. On the whole, the proposed model could be applied without difficulty to handle uncertain core quality with limited data. This approach has potential for core acquisition management in remanufacturing. The results offer considerable managerial insights into remanufacturing practices. Firstly, due to the incompleteness of data in core acquisition management, it is suggested to apply a multi-criteria sorting model with AHP-entropy weights coupled with grey clustering as a short-term quality control method rather than a probability statistics approach. Secondly, in general, the important criteria for the quality sorting of incoming cores are the physical, technical, and usage conditions, which are represented by a new dimension to evaluate the quality of used products. The research has also shown the practical contribution of this research with a case study of a quality sorting problem with a heavy-duty equipment remanufacturer. These results demonstrate potential for the general applicability to remanufacturing. The current study was limited by the independence between the proposed criteria and sub-criteria. Further research needs to examine the dependencies between the proposed criteria and sub-criteria in more detail.

**Author Contributions:** Conceptualization, M.I.M. and U.C.; methodology, M.I.M. and U.C.; software, M.I.M.; validation, M.I.M., U.C. and N.K.; formal analysis, M.I.M. and N.K.; investigation, M.I.M.; resources, U.C. and N.K.; data curation, M.I.M.; writing—original draft preparation, M.I.M.; writing—review and editing, M.I.M., U.C. and N.K.; visualization, M.I.M.; supervision, U.C. and N.K.; project administration, M.I.M.; funding acquisition, U.C. All authors have read and agreed to the published version of the manuscript.

**Funding:** This research was funded by Directorate Research and Community Services (DRPM) Institut Teknologi Sepuluh Nopember (ITS) Surabaya, grant number 930/PKS/ITS/2020.

**Institutional Review Board Statement:** Not applicable.

**Informed Consent Statement:** Not applicable.

**Data Availability Statement:** Not applicable.

**Acknowledgments:** The authors extend their gratitude to DRPM ITS for the funding program from Postgraduate Research Grant of 2020 Dissertation Doctoral Research Grant.

**Conflicts of Interest:** The authors declare no conflict of interest.

# Appendix A

**Table A1.** List of criteria and sub-criteria considered for assessing the quality levels of incoming cores.

| Sub-Criteria | Description | Assessment | Value Range | Reference Value | Target Value |
|---|---|---|---|---|---|
| Technological conditions: | | | | | |
| $C_{11}$: Obsolescence | The condition in which the technology of used product has shown that it is out of date, as the product's life cycle was longer than the design life and the emergence of new technological innovations | Qualitative | 1–5 (scale) | 5 | Max |
| $C_{12}$: Upgradability | The ability of a used product in a remanufacturing process to be upgraded for functional or feature enrichment so that the product may more easily to adapt to new technology to avoid obsolescence | Qualitative | 1–5 (scale) | 5 | Max |
| $C_{13}$: Multiple life cycles | The condition of the used product's life cycle such that it can be recovered for its useful life | Qualitative | 1–5 (scale) | 5 | Max |
| $C_{14}$: Disassemblability | The ability of the product or core regarding the ease of disassembly | Qualitative | 1–5 (scale) | 5 | Max |
| Physical conditions: | | | | | |
| $C_{21}$: Damage level | The degree of physical defects or damage, for example, cracks, corrosion, and wear | Qualitative | 1–5 (scale) | 1 | Min |
| $C_{22}$: Components completeness | The level of completeness of the components as a whole system of used products | $\left( \dfrac{No\ of\ available\ components}{Total\ no.\ of\ components} \right) \times 100\%$ | 0–100 (%) | 0% | Max |
| $C_{23}$: Traceability of identity | Ease of tracing for the product variation information of the model or type, for example, the manufacturer's number is a readable identification number | Qualitative | 1–5 (scale) | 5 | Max |
| $C_{24}$: Geometric and dimensional tolerance | The allowable dimensional or geometrical variation limit | Qualitative | 1–5 (scale) | 1 | Min |
| Usage conditions: | | | | | |
| $C_{31}$: Frequency of uses | The frequency of using the product during the usage phase | Qualitative | 1–5 (scale) | 1 | Min |
| $C_{32}$: Maintenance frequency | The intensity of product maintenance is carried out during the use phase | Qualitative | 1–5 (scale) | 5 | Max |
| $C_{32}$: Remaining useful of life | The remaining usable time for a specified period. | Qualitative | 1–5 (scale) | 5 | Max |

**Table A2.** Assessment example.

| Sub-Criteria | Expert's Assessments | Answer | | |
|---|---|---|---|---|
| | | **Linguistics Value** | **Description** | **Scale** |
| $c_{11}$: Obsolescence | How is the equality of the conditions of the used product technology regarding the emergence of new technological innovations? | Very high | Equal as new technology | 5 |
| | | High | Very good | 4 |
| | | Moderate | Good | 3 |
| | | Low | Acceptable | 2 |
| | | Very low | Overtime | 1 |
| $c_{12}$: Upgradability | What is the ability of the used product to be upgraded to improve the functionality or add features such that new technologies may be more easily adapted? | Very high | Minimal repair | 5 |
| | | High | Imperfect repair | 4 |
| | | Moderate | Replacement with younger parts | 3 |
| | | Low | Complete/perfect repair | 2 |
| | | Very low | Replacement with new parts | 1 |
| $c_{13}$: Multiple lifecycles | How many life cycles of a used product may occur for remanufacturing? | Very high | More than four cycles | 5 |
| | | High | Four cycles | 4 |
| | | Moderate | Three cycles | 3 |
| | | Low | Two cycles | 2 |
| | | Very low | One cycle | 1 |
| $c_{14}$: Disassemblability | How easy is it to disassemble the used product? | Very high | No connections Disassembly will not lead to any damages to the parts; manual operation; quick disassembly | 5 |
| | | High | Non-permanent joints; screws (to be removed) Connections are not destroyed for disassembly; joint disassembly will not lead to any damage to the parts; manual operation; quick disassembly | 4 |
| | | Moderate | Non-permanent snap-fit joints (to be opened); disassembly will not lead to any damage to the parts; manual operation is possible and time-consuming | 3 |
| | | Low | Non-permanent joints; clips (to be removed); disassembly will not lead to any damages to the parts; powered tools are often needed; time consuming | 2 |
| | | Very low | Permanent joints (to be broken; disassembly will lead to damage to the parts; large powered tools are required and such work is time-consuming | 1 |

**Table A2.** *Cont.*

| Sub-Criteria | Expert's Assessments | Answer | | |
| --- | --- | --- | --- | --- |
| | | Linguistics Value | Description | Scale |
| $c_{21}$: Damage level | What is the degree of damage? | Very high | $x \geq 4 \text{ mm}^3$ | 5 |
| | | High | $3 \text{ mm}^3 \leq x < 4 \text{ mm}^3$ | 4 |
| | | Moderate | $2 \text{ mm}^3 \leq x < 3 \text{ mm}^3$ | 3 |
| | | Low | $1 \text{ mm}^3 \leq x < 2 \text{ mm}^3$ | 2 |
| | | Very low | $0 < x < 1 \text{ mm}^3$ | 1 |
| $c_{23}$: Traceability of identity | How is the ability to track the specifications or identity information of used products? | Very high | Radio frequency identification (RFID) | 5 |
| | | High | One-dimensional linear barcode | 4 |
| | | Moderate | Readable text and labels | 3 |
| | | Low | Text or symbols are less readable | 2 |
| | | Very low | Incomplete text or symbols | 1 |
| $c_{24}$: Geometric and dimensional tolerance | Deformation from the ideal shape and dimension within tolerance values? | Very high | Highest deformation from ideal dimension and shape (exceeding the acceptable tolerance) | 5 |
| | | High | Higher deformation from ideal shape and dimension (still acceptable tolerance) | 4 |
| | | Moderate | Moderate deformation from ideal shape and dimension (still acceptable tolerance) | 3 |
| | | Low | Lower deformation from ideal shape and dimension (still acceptable tolerance) | 2 |
| | | Very low | No deformation from ideal shape and dimension | 1 |
| $C_{31}$: Frequency of uses | Frequency of use or operating condition (oc) of the product during the usage phase | Very high | Continuously | 5 |
| | | High | Overload | 4 |
| | | Moderate | Normal load | 3 |
| | | Low | Under load | 2 |
| | | Very low | Occasional | 1 |
| $C_{32}$: Maintenance history | Failure frequency (number of entries in the shutdown register per year) | Very high | Approximately more than 41 times | 5 |
| | | High | Approximately 31–40 times | 4 |
| | | Moderate | Approximately 20–30 times | 3 |
| | | Low | Approximately 11–20 times | 2 |
| | | Very low | Approximately 0–10 times | 1 |
| $C_{33}$: Remaining useful of life | How long is the remaining useful life (RUL) of the incoming core (calculated from the end of the period of use to the end of the useful life)? | Very high | More than four years | 5 |
| | | High | Four years | 4 |
| | | Moderate | Three years | 3 |
| | | Low | Two years | 2 |
| | | Very low | One year | 1 |

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
