# Peer review of "A Novel Multi-Criteria Sorting Model Based on AHP-Entropy Grey Clustering for Dealing with Uncertain Incoming Core Quality in Remanufacturing Systems"

_applsci, doi:10.3390/app11062731_

Round 1
Reviewer 1 Report
This paper has proposed an AHP grey clustering method to evaluate the quality of incoming core for remanufacturing and illustrated application of that method in a case study.
Overall comments
There is a strong need to polish the paper to improve its readability. In its current form, the paper fails to present core ideas and concepts clearly. Also, as review of literature is inadequate, it is unclear how the proposed approach is different from past studies and how it offers higher value in literature and practice.
Specific comments
- Introduction
“The term uncertainty is generally understood to …” on p.2. Need to clearly define the concept of uncertainty in the context of remanufacturing.
Need to clarify the meaning of quality sorting, quality grading and quality classification on p.2. Are they the same?
Need to further clarify long-term and short-term quality control.
Firstly, should present the major challenges and characteristics of the quality sorting problem of the incoming core clearly. Then the weaknesses of current approaches. Finally, describe and explain the approach proposed and adopted in the current study to address the weaknesses.
3.1. Problem structuring
Need to strengthen the literature review. Explain how does the proposed framework presented in Figure 3 offer higher value and address weaknesses of past studies?
3.2. Evaluation processes
Line 357: Need to justify why 3 grey classes (best, middle, or worst quality) are appropriate for the case study.
3.3. Assignment to the classes
To fully demonstrate the value of this research, it is better to present a comparison of results of the proposed and conventional approaches.
Literature Review
The clustering weights could be obtained through different methods (e.g. arithmetic mean, the Shannon Entropy method) and be complemented by the Single Factor method. There is a need to present an in-depth discussion of other approaches in literature review and explain how the proposed approach builds on past studies and offers higher value.
Need to present the limitations of the proposed approach.
Table A1
It is unclear what Reference refers to.
C22: The equation is incomplete.
Table A2
C13: “How many times the ability to a used product life cycle that can be recovered for its useful life>” The meaning of this criterion is unclear.
Reviewer 2 Report
Reading the title and having a first glance at the abstract it looked a promising paper, as it addresses a major energy management topic that has received extensive scientific interest lately, namely the environmental management system sustainable performance. The topic is highly relevant and fit well into the scope of the “Applied Sciences” Journal. The article is very interesting for the readers of the journal, but a number of improvements have to be made before published. In this light, I suggest a minor revision of the article in order to merit publication. I hope that the criticisms I present below in bullet form will help the author improve the paper.
My main objections for publishing the paper in its current form are the following:
- Within the introductory section, the authors should provide clear and detailed analysis of the basic scope of the paper. The authors should elaborate on the problem statement.
- The authors should give a more detailed analysis and clarifications on the methodological advances of the presented approach. I am not completely convinced about the innovation of the methodological approach adopted in the proposed paper. Within this section the authors should provide a flowchart of the adopted methodology in order to illustrate a clear view to the readers. I would suggest the creation of a descriptive flowchart depicting the major steps of the proposed methodology. The exact flow of the methodology framework that have been utilized within the context of this study is not clear.
- In the conclusion’s section, I would expect some more managerial insights and general comments. The authors should clearly reconstruct this section.
Reviewer 3 Report
The paper brings significant contributions to literature and represents a useful resource for practitioners as well. The topic is fresh, and your research is innovative and original. All the theoretical concepts included in the model are well explained, the content is clear, and the paper has a logical structure. However, in the final version of the paper you should explain how many experts were asked to estimate the relative preference of criteria and sub-criteria (after the line 330), what is their professional background, and how they were selected. The conclusions should be extended as well, by including some comments about the research limits. Finally, the revised version of the manuscript should fix some language errors (ex. lines 357-358 “are could be”).
Round 2
Reviewer 1 Report
Add a question mark to each research question in Table 1.
Correct the typo “,” in Table 18. That should be “.”.
Professional editing is suggested before publication of the paper. For example, the following sentences should be refined/revised to enhance the readability.
Lines 604-607: What is striking in this chart is that the phenomenon of the use of the subjective-based based on the AHP pairwise comparison weighting method in the proposed model gives the result that the cores assignment results tend to be dominant in certain classes. Whereas, the use of objective-based on entropy weighting results are more balanced in assigning cores into overall classes.
Lines 616-617: The model working on limited data, is a novel sorting model for solving to quality uncertainty problem in core acquisition management.
Lines 658-659: The most obvious finding to emerge from this study is that the weights in subjective weighting tend to have more vary than objective weighting.
